# Reprogramming Glioblastoma Cells into Non-Cancerous Neuronal Cells as a Novel Anti-Cancer Strategy

**DOI:** 10.3390/cells13110897

**Published:** 2024-05-23

**Authors:** Michael Q. Jiang, Shan Ping Yu, Takira Estaba, Emily Choi, Ken Berglund, Xiaohuan Gu, Ling Wei

**Affiliations:** 1Department of Anesthesiology, Emory University School of Medicine, Atlanta, GA 30033, USA; michaeljiang@emory.edu (M.Q.J.); testaba@emory.edu (T.E.); emily.choi@emory.edu (E.C.); xiaohuan.gu@emory.edu (X.G.); 2Center for Visual and Neurocognitive Rehabilitation, Atlanta Veterans Affairs Medical Center, Decatur, GA 30033, USA; 3Department of Hematology and Oncology, Emory University School of Medicine, Atlanta, GA 30033, USA; 4Department of Neurosurgery, Emory University School of Medicine, Atlanta, GA 30322, USA; ken.berglund@emory.edu

**Keywords:** glioblastoma, direct reprogramming, hypoxia, apoptosis, induced neurons, p53, Wnt-3α, tumor growth

## Abstract

Glioblastoma Multiforme (GBM) is an aggressive brain tumor with a high mortality rate. Direct reprogramming of glial cells to different cell lineages, such as induced neural stem cells (iNSCs) and induced neurons (iNeurons), provides genetic tools to manipulate a cell’s fate as a potential therapy for neurological diseases. NeuroD1 (ND1) is a master transcriptional factor for neurogenesis and it promotes neuronal differentiation. In the present study, we tested the hypothesis that the expression of ND1 in GBM cells can force them to differentiate toward post-mitotic neurons and halt GBM tumor progression. In cultured human GBM cell lines, including LN229, U87, and U373 as temozolomide (TMZ)-sensitive and T98G as TMZ-resistant cells, the neuronal lineage conversion was induced by an adeno-associated virus (AAV) package carrying ND1. Twenty-one days after AAV-ND1 transduction, ND1-expressing cells displayed neuronal markers MAP2, TUJ1, and NeuN. The ND1-induced transdifferentiation was regulated by Wnt signaling and markedly enhanced under a hypoxic condition (2% O_2_ vs. 21% O_2_). ND1-expressing GBM cultures had fewer BrdU-positive proliferating cells compared to vector control cultures. Increased cell death was visualized by TUNEL staining, and reduced migrative activity was demonstrated in the wound-healing test after ND1 reprogramming in both TMZ-sensitive and -resistant GBM cells. In a striking contrast to cancer cells, converted cells expressed the anti-tumor gene p53. In an orthotopical GBM mouse model, AAV-ND1-reprogrammed U373 cells were transplanted into the fornix of the cyclosporine-immunocompromised C57BL/6 mouse brain. Compared to control GBM cell-formed tumors, cells from ND1-reprogrammed cultures formed smaller tumors and expressed neuronal markers such as TUJ1 in the brain. Thus, reprogramming using a single-factor ND1 overcame drug resistance, converting malignant cells of heterogeneous GBM cells to normal neuron-like cells in vitro and in vivo. These novel observations warrant further research using patient-derived GBM cells and patient-derived xenograft (*PDX*) *models* as a potentially effective treatment for a deadly brain cancer and likely other astrocytoma tumors.

## 1. Introduction

Glioblastoma Multiforme (GBM) is a fast-growing tumor in the adult brain and spinal cord. It is the most common type of primary malignant brain tumor derived from glial cells [1,2]. The main subtypes of gliomas are subdivided based on the glial cell type, including ependymomas, astrocytomas, oligodendrogliomas, and mixed gliomas [3]. Specifically, GBM is a type of glioma derived from astrocytes; it has aggressive characteristics and presents as a uniquely life-limiting illness [4]. From its primary location, such as in subcortical white matter and/or deep brain regions, GBM can spread to multiple regions with poor margination, making it a severe challenge for surgical approaches [5,6]. The difficulties in resecting tumor tissues and removing residual glioblastoma stem cells (GSCs), the cells’ resistance to current cancer therapies, and a high recurrence rate are some of the significant barriers to treating GBM [6,7,8]. For example, GBM tumors are resistant to immunotherapy. PD-L1 inhibitors are of limited efficacy in treating GBM tumors [9]. Other therapeutic interventions include chemotherapies of bevacizumab and temozolomide (TMZ) [10,11,12]. Bevacizumab is a monoclonal antibody that binds to VEGF and inhibits the vascular growth of tumor blood vessels. Unfortunately, a recent phase III trial with bevacizumab revealed greater deteriorations in neurocognitive function, disease symptoms, quality of life, and patient-reported outcomes [13]. TMZ is a DNA alkylating agent and a standard treatment for GBM [14]. However, TMZ shows extensive side effects, including hepatotoxicity and severe hematologic toxicity [15,16]. Meanwhile, GBM cells often exhibit resistance to TMZ treatments [17]. Due to high malignancy and the lack of effective therapies, the median survival of GBM patients ranges from only 5 to 24 months [18,19,20]. The aggressive nature and challenges associated with treating GBM and other astrocytoma tumors merit the development of innovative strategies with distinct mechanisms in basic and translational research.

Novel therapies aimed at combating GBM must account for the specific phenotypic characteristics of these brain tumors [21]. Non-surgical genetic approaches that can alter the inherited cancerous properties of GBM cells have become possible thanks to recent conceptual and technical breakthroughs. An attractive strategy is cell transdifferentiation using transcriptional factors [22,23,24]. Soon after the groundbreaking generation of induced pluripotent stem cells (iPSCs) from somatic cells such as skin fibroblasts [25,26], Kim et al. and others reported and developed direct reprogramming of fibroblasts. In this strategy, multiple transcription factors, e.g., Oct4, Sox2, Klf4, and c-Myc, can reprogram cells to bypass the stemness stage and directly become different lineage cells such as induced neural stem/progenitor cells (iNSCs/iNPCs) and induced neurons (iNeurons) [27,28]. Several subsequent investigations showed improved methods of direct reprogramming of different cells. For example, human fibroblasts and blood cells were directly converted to neural progenitor cells (iNPCs) using a single transcription factor OCT4. Converted iNPCs possess the potential to give rise to all three major subtypes of neural cells: astrocytes, oligodendrocytes, and neurons, all of which are presumably useful for disease modeling and basic studies of early neural fate induction [29,30]. Moreover, transdifferentiation provides a novel and more efficient genetic method of manipulating cell lineages for the treatment of brain injuries and neurodegenerative diseases [22,23,24].

Because of concerns over the preserved proliferating ability of transformed cells, the recent research trend is to focus on transforming glial lineage cells into terminally differentiated cells, such as iNeurons that lack the ability to replicate [31,32,33]. A recent report showed that human GBM U87MG cells could be reprogrammed into terminally differentiated neurons by using a small molecule cocktail comprising forskolin, *N*-Cyclopropyl-5-(2-thienyl)-3-isoxazolecarboxamide (ISX9), CHIR99021 I-BET 151, and (2*S*)-*N*-[(3,5-Difluorophenyl)acetyl]-L-alanyl-2-phenyl-glycine 1,1-dimethylethyl ester (DAPT). These reprogrammed cells displayed neuronal morphological and immunofluorescent phenotypes with decreased viability and a reduced ability to form tumor-like spheroids in vitro [32].

NeuroD1 (ND1) is a master transcriptional factor for neurogenesis with critical roles in regulating neuronal differentiation and maturation during nervous system development [34,35]. We previously investigated ND1-induced direct reprogramming of astrocytes in vitro and after ischemic stroke in the mouse brain [36]. It was identified that virus-enforced ND1 expression alone sufficiently converted reactive astrocytes to functional mature neurons [36]. In the post-stroke brain, ND1 expression in the peri-infarct region significantly reduced astrogliosis and promoted structural and functional recovery [36]. In the present investigation, we tested the hypothesis that the expression of ND1 can redirect the fate of malignant GBM cells of astrocyte origin to post-mitotic neuronal cells and ultimately slow down or halt GBM tumor progression. This novel anti-GBM strategy shows some unique and superior advantages over conventional cancer therapies, and the converted cells may be further explored as a cellular resource in brain repair for tumor-related damages.

## 2. Methods and Materials

### 2.1. GBM Cell Lines

Selected human GBM cell lines consisting of primarily malignant astrocytes including the TMZ-sensitive lines LN229, U373, and U87 and the TMZ-resistant cell line T98G were tested in this investigation. Cell lines were from the Cellular and Immunotherapy Core of Winship Cancer Institute of Emory University, which were originally obtained from the American Type Culture Collection (ATCC, Manassas, VA, USA).

### 2.2. GBM Cell Cultures

GBM cells were cultured using N2/B27 supplemented serum-free medium with 21% O_2_, 5% CO_2_, at 37 °C. The medium was prepared with 45% Dulbecco’s modified eagle medium: nutrient mixture F-12 (DMEM/F12; Sigma-Aldrich, St Louis, MO, USA), 45% Neurobasal (Thermo Fisher Scientific, Waltham, MA, USA), 0.5% N2 supplement (Thermo Fisher Scientific), 1% B27 supplement (Thermo Fisher Scientific), 1% GlutaMAX, 1% nonessential amino acids (Sigma-Aldrich), 0.1 mM β-mercaptoethanol (β-ME; Sigma-Aldrich), 100 U/mL penicillin/streptomycin (Sigma-Aldrich), 5% knockout serum replacement (KSR; Thermo Fisher Scientific). After three passages, GMB cell lines were infected using AAV9-ND1 to induce cell lineage reprogramming for up to 21 days. Cell cultures were randomly divided between the control and ND1 expression groups for in vitro experiments.

For control astrocyte cell cultures, cells were dissected from the postnatal day 7 (P7) mouse brain using mechanical dissociation and cultured under the regular condition using N2/B27 supplemented serum-free medium with 21% O_2_, 5% CO_2_, at 37 °C.

### 2.3. Viral Transduction of GBM Cell Line Cultures and Generation of AAV Vector Packages

AAV-mediated gene delivery has been an efficient and relatively safe tool for gene therapy in treating various brain diseases [36,37]. Astrocytes-derived astrocytomas are the cellular basis of GBM tumors. Genetically targeting astrocytes has been achieved by utilizing the *glial fibrillary acidic protein* (*GFAP*) promoter. We constructed an AAV9 using the human GFAP and ND1 sequences tagged with a reporter gene mCherry or GFP. The mCherry tag, which was ligated to FUGW using two-step overlap PCR to form the control vector AAV9 GFAP-IRES-mCherry (AAV-mCherry) and the ND1 vector AAV9 GFAP-ND1-IRES-mCherry (AAV-ND1). A purified ND1 fragment was ligated into the FUGW plasmid. Verification of correct ligation and plasmid generation was confirmed using PCR and DNA sequence analysis. Plasmid production utilized Stbl3 bacteria, and DNA was purified using Qiagen Miniprep and Maxiprep kits. Viral titer 6.4 × 10^12^ was obtained via two rounds ultra centrifugation from 144 mL to 1.2 mL to 30 µL. Titer calculations were assayed in triplicate using HEK 293FT cell cultures using 1:10 and 1:100 dilutions of concentrated virus, which were fixed and stained for mCherry.

Human GBM cell lines expand quickly, with doubling times of under 24 h. As a control, we used an identical virus expressing mCherry alone without ND1. Within just 2 days, high levels of mCherry could be observed in approximately 80% of cells. The levels increased to 100% within 4 days. We observed that the efficiency of viral expression and reprogramming depended on the initial seeding density of cell cultures. There was an inverse linear relationship between the seeding density and the reprogramming efficacy. Lower-density cultures expressed the highest levels of the reporter gene mCherry two days after infection, and the higher-density cultures expressed lower levels of the reporter gene at the same time point.

Experiments were performed in triplicate at a minimum. At least 10 samples were acquired during the cloning steps, and 3 bacterial strains were sequenced to isolate a mutation-free strain. In transfection experiments, 3 wells of cells were examined for efficacy. DNA concentrations were assessed using a Gen5 ultraviolet spectrophotometry by BioTek (Winooski, VT, USA).

### 2.4. Western Blotting Analysis

Cultured U87 whole cell lysates were collected, and the protein concentration of each sample was determined using the Bicinchoninic Acid Assay (Sigma-Aldrich). We prepared 15% SDS-PAGE gels and loaded them with 20 µm of protein per sample using the Bio-Rad Mini-PROTEAN Tetra Handcast system (Bio-Rad, Hercules, CA, USA). The proteins were transferred to a PVDF membrane (Bio-Rad) using the Bio-Rad Mini Trans-Blot Cell System (Bio-Rad). Membranes were blocked with buffer (5.0% BSA solution) at room temperature for 4 h. Membranes were then incubated overnight at 4 °C with rabbit polyclonal antibody to p53 (1:1000, Cell Signaling Technology, Danvers, MA, USA), BDNF receptor (1:1000, Cell Signaling Technology), Cleaved Caspase-3 (1:1000, Cell Signaling Technology), MAP2 (1:1000, Cell Signaling Technology), β-Catenin (1:1000, Cell Signaling Technology), and ND1 (1:1000, Abcam, Boston, MA, USA). A mouse monoclonal antibody to β-actin (1:1000, Cell Signaling Technology) and a rabbit monoclonal antibody to β-tubulin (1:1000, Cell Signaling Technology) were used as the loading controls. The blots were washed in 0.5% TBS-T and incubated with alkaline phosphatase-conjugated goat anti-rabbit IgG or donkey anti-mouse IgG (Promega, Madison, WI, USA) overnight at 4 °C. Membranes were washed with TBS-T followed by three washes with TBS. Finally, protein concentrations were detected using the addition of BCIP/NBT solution (Sigma-Aldrich).

### 2.5. MTT Assay

The MTT (3-(4,5-dimethylthiazol-2-yl)-2,5-diphenyltetrazolium bromide) assay was employed to assess tumor cell death following AAV-mCherry and AAV-ND1 infection reprogramming in an in vitro GBM model. The viability of tumor cells was evaluated using the MTT assay after 14 days of reprogramming. This colorimetric assay relies on the reduction of the yellow MTT dye to formazan crystals by metabolically active cells. The formazan crystals formed were solubilized, and the absorbance was detected spectrophotometrically. The extent of formazan production directly correlates with the number of viable cells present in the culture, quantifying the differences in cell viability. Cell viability post-viral reprogramming treatment was compared to mCherry-only controls to evaluate the efficacy of the reprogramming strategy in inducing tumor cell death. The MTT assay provides evidence of the cytotoxic effects of viral reprogramming on tumor cells in vitro, providing an additional therapeutic benefit of using reprogramming in cancer treatment.

### 2.6. Wound-Healing Migration Assay

Cell migration ability was assessed using a wound-healing assay. U373 and T98G GBM cells were cultured in a 6-well plate and infected with AAV-mCherry or AAV-ND1. ND1 expression was evaluated 24 h later, and the cells then proliferated for an additional 7 days. A scratch test assay was completed on day 10, when the cultures were scratched with a 200-μL pipette tip along the central vertical line. Scratch debris was removed by washing the cells in PBS, and fresh culture medium was added to the plates. Cells were photographed immediately after the scratch and then incubated at 37 °C. Cell migration was monitored and photographed every 12 h for a total of 48 h. The distance the cells migrated into the scratch area was analyzed using the ImageJ software (Version 1.52; National Institute of Health, Bethesda, MD, USA). Migration ability was quantified by the number of cells invading the wounded area.

### 2.7. Immunoflourescent Staining In Vitro

Immunocytochemical staining was completed in control, ND1-treated cell culture, and in vivo transplanted tumors. U87, U373, T98G, and LN229 cells were plated on glass-bottom dishes and treated in normoxia or hypoxia treatment conditions. Dishes were fixed in a 10% formalin-PBS solution for 10 min twice. Cultures were washed 3 times in PBS, treated with 0.2% Triton-X for 5 min, washed 3 additional times in PBS for 5 min, and blocked in 2.0% fish gelatin (Sigma-Aldrich, Burlington, MA, USA) for 1.5 h at room temperature. Slides were incubated overnight at 4 °C in primary antibodies to TUJ1 (1:2000, Abcam, Boston, MA, USA), NeuN (1:400, Abcam, Boston, MA, USA), BrdU (1:500, Abcam), mCherry (1:500, Abcam), Synapsin-1 (1:200, Cell Signaling Technologies, Danvers, MA, USA), Neurofilament (1:500, Chemicon, Temecula, CA, USA), ND1 (1:500, Abcam), or chicken primary antibody to GFAP (1:1000, Invitrogen, Carlsbad, CA, USA). Cultures were then washed 3 times in PBS and treated with Alexa Fluor 488 G goat anti-rabbit IgG (1:1000, Invitrogen), Cy3-conjugated donkey anti-mouse IgG (1:800, Jackson ImmunoResearch, West Grove, PA, USA), Cy5 donkey anti-rabbit IgG (1:400, Invitrogen), or Alexa Fluor 488 goat anti-chicken IgG (1:1000, Invitrogen) for 2 h at room temperature. After an additional 3 PBS washes, dishes were mounted, cover-slipped, and visualized using a fluorescent microscope BX51 (Olympus, Tokyo, Japan).

### 2.8. Immunoflourescent Staining Ex Vivo

For ex vivo immunohistochemical staining, fresh-frozen 10 μm-thick sections were cut on a cryostat microtome (Vibratome 5040, St. Louis, MO, USA). Slides were fixed for 10 min in 10% formalin. Slides were then washed in PBS 3 times for 5 min and then placed in a 2:1 ethanol and glacial acetic acid (respectively) solution for 7 min. Slides were again washed 3 times in PBS and then blocked in 2.0% fish gelatin (Sigma-Aldrich) for 2 h. Slides were then washed 3 times in PBS and treated with primary antibodies IBA-1 (1:500, FujiFilm Wako Pure Chemical Corporation, Osaka, Japan), mCherry (1:500, Abcam), TUJ1 (1:2000, Abcam), ND1 (1:500, Abcam), or GFAP (1:1000, Invitrogen) overnight at 4 °C. Slides were washed in PBS and then treated with Alexa Fluor 488 goat anti-chicken IgG (1:1000, Invitrogen), Cy3-conjugated donkey anti-mouse IgG (1:800, Jackson ImmunoResearch, West Grove, PA, USA), or Cy5 donkey anti-rabbit IgG (1:400, Invitrogen) for 2 h. After an additional 3 PBS washes, slides were mounted using VECTASHIELD (Vector Laboratories, Newark, CA, USA), cover-slipped, and visualized using fluorescent microscope BX51 (Olympus).

### 2.9. TUNEL Assay

Cell death and DNA damage were measured using terminal teoxynucleotidyltransferase-mediated DUTP-biotin nick end labeling (TUNEL) staining. The staining was performed according to the DeadEnd^TM^ Fluorometric TUNEL detection system (G3250; Promega, Madison, WI, USA). Briefly, cells were equilibrated in TdT reaction buffer for 10 min. Cells were then incubated in a TdT incubation cocktail for 60 min at 37 °C, followed by incubation in 2× SSC for 15 min at room temperature and Hoechst 33342 (1:20,000) staining for 5 min. Dishes were mounted using VECTASHIELD Mounting Medium (Vector Laboratories, Newark, CA, USA). TUNEL-positive cells were counted in five non-overlapping fields imaged using a fluorescent microscope BX51 (Olympus), and the ratio of positive cells to total cells was calculated.

### 2.10. Orthotopical GBM Mouse Model

Adult C57/BL mice (3 months old, male) were used to generate an in vivo GBM model by implanting GBM cells into the white matter of the right hemisphere. Although GBM tumors can arise anywhere within the brain, they have a predilection for the subcortical white matter and deep gray matter of the cerebral hemispheres [38]. We therefore selected the fimbria-fornix white matter location as the implantation site. The anatomical and morphological nature of the non-neuronal cells that dominate in this region also favored the identification of converted neuronal cells.

Stereotaxic injection of the control and the ND1 adenovirus was performed using a 10 μL Hamilton GASTIGHT™ syringe (Hamilton Company, Reno, NV, USA). The injection location was into the fimbria-fornix region adjacent to the right lateral ventricle (Stereotaxic coordinates: −1.5 mm posterior (AP), +2.2 ML, and −4.5 DV). A total of 2 µL containing 200,000 cells was injected over 15 min. Transplanted GBM cells could be observed as a solid tumor measuring up to 2 mm in diameter at the injection site. Additional smaller secondary tumors were observed in two of five animals in the control treatment group.

Twenty-one days following the transplantation of the GMB cells, animals were sacrificed for the assessment of tumor formation. Hemoxylin-eosin staining was used to reveal the location of tumors in the fimbria-fornix. Brains were removed and placed in a brain matrix and then sliced into 1-mm coronal sections. Slices were incubated in 2% TTC solution at 37 °C for 5 min and then stored in 10% buffered formalin for 24 h. Digital images of the caudal aspect of each slice were obtained by a flatbed scanner. The infarct, ipsilateral hemisphere, and contralateral hemisphere areas were measured using ImageJ software (NIH). The indirect method (subtraction of residual right hemisphere cortical volume from cortical volume of the intact left hemisphere) was used for infarct volume calculation. Infarct measurements were performed under double-blind conditions.

### 2.11. H&E Staining of Ex Vivo Histology

Hematoxylin and eosin (H&E) staining was employed to histologically examine mouse brain sections for the identification of tumors in the fimbria. Cellular morphology and tissue architecture were visualized under a light microscope. Tumors were identified based on characteristic histological features such as abnormal cellularity, cell density, nuclear pleomorphism, increased mitotic activity, and disruption of normal tissue architecture in the white matter background of the fimbria. A thorough examination of H&E-stained sections spanning the injection location as well as surrounding tissues facilitated the accurate classification and characterization of brain tumors as small as 100 μm across, ensuring that we did not miss any smaller secondary tumors outside of the injection location.

### 2.12. Statistical Analysis

The Student’s unpaired *t* test was used for analysis between two experimental groups. One-way ANOVA analysis followed by Tukey’s post hoc tests was performed for multiple pair-wise comparisons. Statistical analysis and graph representations were completed using the GraphPad Prism software version 9 (Dotmatics, Boston, MA, USA). Differences were considered significant at *p* < 0.05.

## 3. Results

### 3.1. Expression of ND1 in GBM Cells and Neuronal Reprogramming

Our previous investigations using an AAV-ND1 package effectively reprogrammed reactive astrocytes to iNeurons [36]. In the present investigation, mCherry or eGFP-tagged AAV-ND1 packages were generated for in vitro and in vivo reprogramming of GBM cells. The characterization of our reprogramming viruses has previously been reported [36]. GBM cell lines tested in the current study included TMZ-sensitive cells (LN229, U373, and U87) and TMZ-resistant cell line T98G. Cell cultures were infected with the control AAV-mCherry vector or the AAV-ND1-mCherry (AAV-ND1) vector, then cultured for up to 28 days post-infection to examine the efficacy and efficiency of ND1 expression, cell proliferation, cell death, and neuronal conversion. We verified that our control vector and AAV-ND1 vector similarly infected GMB cells, which ensured fair comparisons in the following experiments (Figure 1A).

The expression levels of ND1 were inspected using Western blotting in cultured human GBM cells as well as in normal cells from the postnatal day 7 (P7) mouse brain. As expected, a high level of endogenous ND1 expression was identified in normal cells while very low levels of ND1 were seen in naïve GBM cells (Figure 1B). Meanwhile, the cell motility/cycle regulatory gene β-catenin in the Wnt/β-catenin axis [39,40] was universally very low in GBM cells (Figure 1B). Although the Wnt/β-catenin axis is usually regarded as an oncogenic pathway [41,42], its activation promotes neuronal differentiation of GBM cells [43]. Substantially low levels of ND1 and β-catenin in GBM cell lines suggested that the manipulation of ND1 expression and the Wnt/β-catenin pathway could significantly impact phenotypical fate and cellular activities.

As the first step of testing our hypothesis, AAV-ND1 was added into the culture medium to induce the expression of ND1 in several GBM cell lines. Around 3–14 days after transduction, the ND1 level was enhanced in infected GBM cells (Figure 1C). Seven to 14 days after ND1 transduction, immunocytochemical staining using the neuronal lineage markers Tuj1 verified a lineage switch to neuronal cells (Figure 1D). The early neuronal marker TUJ-1 emerged from 7 to 14 days post-infection, while mature neuronal marker NeuN was detected 21–28 days post-infection (Figure 1D). Consistent with this conversion, the Western blot analysis revealed the expression of the neurogenesis and neuronal marker microtubule associated protein 2 (MAP2) (Figure 1F,G). Brain-derived neurotrophic factor (BDNF) plays an important role in the growth and differentiation of neurons and synapses; the BDNF expression was essentially absent in naïve GBM cells, while the ND1 conversion significantly increased the BNDF expression (Figure 1F,G).

A main characteristic of cancer cells, including the cell lines tested in this investigation, is the absence of the anti-cancer gene p53. Captivatingly, ND1-converted cells not only expressed neuronal lineage markers but also acquired significant p53 expression. The acquired p53 expression reached a level close to that in normal cells (Figure 1G). Consistent with the expectation that post-mitotic cells such as neurons are more vulnerable to programmed death [44], ND1 reprogramming significantly increased TUNEL-positive cells—an indication of increased DNA damage and apoptotic cell death (Figure 1H). As a result, surviving GBM cells significantly decreased in ND1-reprogrammed cultures (Figure 1I).

### 3.2. Decreased Proliferation of GBM Cells after ND1 Reprogramming

To verify that ND1 expression attenuated the proliferation of GBM cells, the proliferation marker BrdU was measured on different days after transduction. In both TMZ-sensitive and -resistant GBM cells subjected to AAV-ND1 transduction, BrdU-positive cells were significantly reduced compared to cells of vector controls (Figure 2). Three days after viral transfection, there were 50% or more reductions of BrdU-positive GBM cells in LN229, U87 and T98G cultures in immunostaining experiments (Figure 2).

### 3.3. Promoting Effect of Hypoxia on ND1-Induced Neuronal Conversion of GBM Cells

Hypoxic conditions exist inside solid tumors, considerably altering the local microenvironment, gene expression, and neurovascular infrastructures [45]. We thus examined ND1-induced GBM cell reprogramming at a low oxygen level of 2% O_2_ versus the regular 21% O_2_ widely used in cell cultures. The hypoxic environment of 2% O_2_ promoted cell proliferation in the control vector group, while the increased proliferation was not seen in ND1-expressing cells (Figure 3A). In fact, proliferation was significantly reduced compared to vector control cells (Figure 3A).

The hypoxic condition showed a facilitating effect on neuronal maturation as more converted cells expressed the mature neuronal markers TUJ-1, NeuN, and synaptic protein synaptin-1 (Figure 3B–E). Converted cells under 2% O_2_ also underwent more extensive morphological transformations such as enlarged axonal extension (Figure 3). Meanwhile, hypoxia markedly enhanced the death of converted cells compared to the cell death under 21% O_2_ (Figure 3F).

### 3.4. Attenuated Cell Migration of ND1-Reprogrammed Cells

Cancer cells, especially GBM cells, behave with aggressive motility compared to normal cells [46]. The wound-healing assay was performed in U87 and T98G cell cultures 7 days after ND1 transduction by creating a wounded area across the culture dish. The empty area was gradually covered by migrating neighboring cells in the next 24–36 h (Figure 4A). In GBM cultures treated with AVV-ND1 transduction, migration was slower and significantly fewer ND1-expressing cells appeared in the empty area compared to the migrating cells in the control group (Figure 4B,C).

### 3.5. Regulatory Roles of the Wnt-3 Pathway in ND1-Induced Reprogramming and Neuronal Conversion of GBM Cells

Hypoxia selectively activates the Wnt signaling pathway in cancer cells, which mediates many downstream effects of hypoxia [47,48]. On the other hand, hypoxia stimulates neurogenesis of human neural stem cells via the activation of the Wnt pathway [48]. To test if the Wnt signaling was a mediator or promoter in ND1-induced reprogramming, manipulations of the Wnt/β catenin pathway were performed with AAV-ND1 and control AAV transduction. Seven days after transduction, BrdU-positive cells in the presence and absence of Wnt-3α (100 nM) or the Wnt/β catenin inhibitor XAV939 (1–20 ng/mL) were inspected using fluorescent imaging. In LN229 cultures, both Wnt 3α and XAV939 markedly reduced BrdU-positive cells in vector control cells (Figure 5A). In LN229 cultures transfected with AAV-ND1, proliferating BrdU-positive cells were generally low as reported above, and neither Wnt 3α nor XAV939 showed any further influence (Figure 5A).

T98G cells had different reactions to Wnt/β-catenin regulations. In the control vector cultures, inhibition of the Wnt signaling significantly decreased the number of BrdU-positive cells. However, in AAV-ND1 transduced cultures, Wnt 3α significantly increased proliferation while XAV939 showed an inhibitory effect (Figure 5A). While reactions to Wnt3α and Wnt inhibition in T98G cell cultures were consistent with the common understanding of the Wnt signaling in GBM proliferation, the reaction of LN229 cells was unexpected in both vector control and ND1 groups. Cellular and gene heterogeneities exist among different GBM cell lines, and the Wnt signaling plays significant roles in both neuronal differentiation and proliferation, which are two cellular processes that are functionally opposed to each other. Therefore, the contradictory data might be a result of the interactions of multiple regulations, although the detailed mechanism is not well understood. It is likely that the balance of these regulations, which involves the canonical/β-catenin-dependent and the non-canonical pathways, jointly determines the ultimate outcome of Wnt signaling changes in transdifferentiation of different GBM cells.

Additional immunocytochemical assessments revealed some drastic effects of Wnt 3α on cellular markers associated with ND1 reprogramming of LN229 cells. The addition of Wnt 3α (100 nM) displayed a huge inhibitory action on the ND1 expression in GBM cells with either the control vector or AVV-ND1 (Figure 5B). Interestingly, Wnt 3α largely diminished the reactive astrocyte marker GFAP, which might, at least partly, account for the reduced ND1 transduction/expression, since the GFAP promoter was used in the infection process (Figure 5C). Along with this inhibitory effect, the addition of Wnt 3α predominantly prevented ND1-induced neuronal conversion, as evidenced by significant reductions in neural and neuronal gene expressions of Nestin, MAP2, and TUJ-1 (Figure 5D–F). Consistent with these assessments, the transcription factor and astrocyte/neuron developmental marker PAX6 were markedly blocked in the presence of Wnt 3α (Figure 5G). According to previous investigations, this PAX6 suppression may be another molecular mechanism for the restrained GBM tumorigenicity and invasiveness [49,50].

### 3.6. ND1 Reprogramming Therapy in a GBM Mouse Model

The therapeutic effect of ND1 reprogramming therapy was tested in a mouse model established by the transplantation of GBM cells transfected with AAV-mCherry or AAV-mCherry-ND1. Three days after viral transduction, the control GBM cells and ND1-reprogrammed cells (200,000 cells in 2 μL media) were implanted into the white matter of the right hemisphere of an adult mouse, accompanied by daily injections of the immunosuppressive reagent cyclosporine (15 mg/kg/day or 0.375 mg/day). Fourteen days later, a sizable tumor was detected in the injection area and satellite tumors could sometimes be seen in adjacent brain regions of mice that received the control GBM cells (Figure 6A). In contrast, mice that received ND1-expressing GBM cells developed a much smaller single tumor in the brain (Figure 6A,B), and no tumor cells/tissues at all were detected in some animals (Figure 6B).

Immunohistochemical imaging showed the absence of the virus reporter mCherry in the contralateral hemisphere, while significant mCherry fluorescence, GFAP-positive cells and TUJ-1 expression of transplanted/converted cells were observed in the implanted region 14 days later (Figure 6C). In the tumor area, there were noticeable IBA-1-positive microglia/macrophage cells accompanied by implanted cells expressing TUJ-1 (Figure 6E). Quantified analysis confirmed that there were significantly fewer GFAP-positive astrocytes and IBA-1-positive microglia cells in the tumors formed by ND1-GBM cells (Figure 6F). On the other hand, abundant TUJ-1 expression was detected in ND1-GBM-formed tumors but not in the control tumors (Figure 6F).

## 4. Discussion

The present investigation provides novel evidence that direct reprogramming using a single transcriptional factor ND1 can effectively convert highly malignant GBM cells into non-cancerous neuronal lineage cells. Converted cells express the neuronal markers TUJ-1, MAP2, NeuN, and synapsin-1, as well as the p53 gene, which is only perceptible in normal cells. Supporting our hypothesis, converted GBM cells demonstrated reduced proliferation, increased cell death, attenuated migration activity, expression of normal cell markers, sensitivity to Wnt signaling, and diminished tumor formation. The cell transdifferentiation strategy using ND1, which has been demonstrated in multiple GBM cell lines, is fundamentally different from most conventional anti-cancer therapies in several ways. The reprogramming strategy does not directly target cell death pathways, instead, the increased death is achieved via cell phenotype manipulation. It shows the unique advantages of enhanced efficacy under a hypoxic condition that usually depreciates the efficacy of current chemical and radiation therapies. In addition, converted cells with normal cellular features may be leveraged for reconstruction of damaged brain tissues caused by tumor formation, surgical removal, and cellular or network deteriorations. This attractive idea will require further specific investigations, as discussed below. Specifically, whether the cell conversion triggers chronic inflammation, their long-term survival, interactions with other cells, and impacts on host neural networks should be delineated for the safety of translational applications.

Being a potential anti-GBM therapy, the goal of the current investigation is different from reprogramming strategies for brain injuries. Whether the converted cells become functional neurons was not a critical criterion or the initial purpose of the anti-cancer strategy. Although ND1-converted GBM cells express the mature neuronal marker NeuN with some typical neuronal morphology, such as elongated axons, these features are less pronounced in the in vivo observation. In contrast to the neurons converted from the normal glial cultures and astrocytes of the stroke brain, our preliminary electrophysiological patch-clamp recordings on converted GBM cells failed to trigger action potentials. Nevertheless, these converted cells express synaptic proteins and may form synaptic connections with existing neurons, although their neuronal functionality could be limited. On the one hand, the conversion of GBM cells into non-functional neurons could be an advantage in cancer therapy, since it avoids the side effects of incorrectly formed neuronal connections and aberrant activities after reprogramming. Future research, however, may explore whether GBM cells can be fully transdifferentiated into mature and functional normal neurons that are useful for repairing tumor-associated brain damage, including those caused by the adverse effects of chemotherapy and radiotherapy.

Viral transfection/transduction has been used to introduce transcription factors and small molecules into cells in transdifferentiation or direct reprogramming strategies [24,32]. The terms transdifferentiation and direct reprogramming or direct conversion have also been used in contexts where one type of stem cell or glial cell is converted into a completely different type of stem cell or glial cell [33,51]. Previous studies showed that transformation agents such as histone deacetylase inhibitors, multi-kinase inhibitors, and bone morphogenetic proteins (BMPs) can revert GBM cells to their glial origins or to other glial cell types such as oligodendrocytes [32,52,53,54,55,56]. Reprogramming was applied to convert GBM cells into astrocytes and mesenchymal cells [52,57,58]. Alternatively, GBM cell-derived iNSCs were utilized to deliver the anti-cancer molecule TRAIL, resulting in a decreased growth of established solid and diffused patient-derived orthotopic xenografts and significantly prolonged median survival of GBM mice [59].

Viral vectors have been tested in experimental gene therapies for neurological disorders, including GBM animal models and human patients, and showed no significant side effects [37,60]. Using lentivirus and AAVs, we previously investigated inter- and intra-lineage reprogramming of terminally differentiated cells into other cell types for neuronal cell replacement following neurological diseases, including ischemic stroke and traumatic brain injury (TBI) [36,61,62]. We showed that via ND1-induced reprogramming of reactive astrocytes in the post-injury brain, cells can be forced to dedifferentiate into mature and functional neurons and show functional benefits [36]. Based on the progress from our and other groups, we identified ND1 as an ideal candidate to investigate how redirecting cell fates can mediate efficient, single-factor reprogramming of malignant glia into terminally differentiated and non-cancerous cells. This approach requires delivery of the viral package to the brain region where the GBM is located, which can be challenging but achievable by intracranial injections and during resectioning of GBM tissues. GSCs are believed to be responsible for drug resistance and tumor recurrence [63,64,65]. With the GFAP promoter, the ND1 transduction can reprogram GSCs that are GFAP-positive cells. Since the promoter of the viral package, the location, and the amount of virus injection can be well controlled, off-target actions on normal brain regions/cells and short- or long-term side effects could be minimized to lower or no significant levels for a life-saving intervention.

Hypoxia is a hallmark of solid tumors that promotes metastasis and represents a significant obstacle to successful cancer therapies [66,67]. In response to hypoxia, cancer cells activate a transcriptional program involving hypoxia-inducible factor 1 (HIF-1) that allows them to survive and thrive in this harsh microenvironment [68]. Growing evidence suggests that the cellular response to hypoxia is more complex than HIF-1 overexpression. Specifically, the hypoxic GBM microenvironment is a key regulator of GSCs and helps maintain tumor cell stemness and prevents differentiation [69,70]. Here, we report that a hypoxic condition mimicking the microenvironment of solid tumors substantially promotes ND1-induced GBM reprogramming and dedifferentiation to neuronal cells. In this process, low O_2_ facilitates ND1 expression, neuronal conversion, and maturation. Meanwhile, similar cell death/proliferation of ND1-converted cells was seen with 21% and 2% O_2_, supporting the assertion that the converted cells in the post-mitotic state were insensitive to the proliferating effect stimulated by hypoxia. This observation is clinically significant, implying that, unlike conventional cancer therapies, a hypoxic condition facilitates rather than hinders the efficacy of reprogramming therapies against solid tumors.

One of the encouraging observations of ND1-induced GBM reprogramming is that the converted cells express the normal cellular protein p53. Up to now, the p53 protein has been regarded as a negative factor in reprogramming approaches because it intrinsically induces apoptosis and cell cycle arrest, which reduces the efficacy and efficiency of generating new cells [71,72]. In basic research of reprogramming somatic cells to iPSCs, some even proposed to knock down p53 to improve the reprogramming efficiency [73]. This concern, however, is entirely reversed for cancer therapy, since the increased cell death and the decreased number of tumor cells are the exact goals of this type of treatment.

The Wnt family of glycoproteins are regulators of embryonic and neural stem cell development, cellular polarity, cell proliferation, and stemness phenotype [63]. On the other hand, the Wnt/β-catenin axis is widely regarded as an oncogenic signaling pathway [74,75,76]. The binding of Wnt proteins to the frizzled receptor and low-density lipoprotein receptor-related proteins triggers either canonical or non-canonical signaling cascades, resulting in β-catenin-dependent and -independent regulations of multiple cellular and molecular mechanisms such as gene expression, stem cell renewal, differentiation, cell proliferation/polarity and Ca^2+^ hemostasis [63]. Aberrant hyperactivation of the Wnt signaling pathway is implicated in tumor growth, recurrence, and self-renewal of GSCs [64]. The Wnt pathway has also been implicated in GBM resistance to TMZ and radiotherapy [65]. Ironically, activation of the Wnt pathway promotes neuronal differentiation of GBM cells [43]. Considering the heterogeneity of GBM cell lines and the multiple, sometimes opposing, functional roles of Wnt signaling pathways, it may not be a surprise that different GBM cell lines responded to Wnt regulation differently. For example, naïve LN229 cells, but not ND1-converted cells, showed reduced proliferation under either an increase or a decrease in Wnt signaling. T98G cells, on the other hand, are more sensitive to the Wnt inhibitor XAV939, showing reduced proliferation either in a naïve or converted state. The sensitivity of reprogramming outcomes to Wnt 3α and XAV939 suggests that Wnt pathways play complicated and unknown regulatory roles in the reprogramming and transdifferentiation of GBM cells.

With ND1 expression, the size, growth, and metastasis of GBM tumors orthotopically transplanted into the white matter of adult mouse brains were significantly reduced. This study, alongside others reporting on combinations of transcription factors, suggests that the direct reprogramming of cancer cells could be a promising therapy for the treatment of GBM. A recent consensus is that conventional human GBM cell lines have undergone multiple genetic alterations, making them unreliable cellular sources for translational GBM research [77]. Alternatively, freshly isolated patient-derived GBM cells provide more suitable tools for translational research. The present investigation is a proof-of-concept study. As one of the first studies on the direct reprogramming of GBM cells, we successfully demonstrated the feasibility and efficacy of a single transcription factor in multiple GBM cell lines. These data warrant further investigations to verify and improve the direct reprogramming strategy in patient-derived GBM/stem cells, preferable in 3-D cell cultures [78,79], and patient-derived xenograft (*PDX*) models that are generated using cells more analogous to clinical cases [80,81].

## Figures and Tables

**Figure 1 cells-13-00897-f001:**
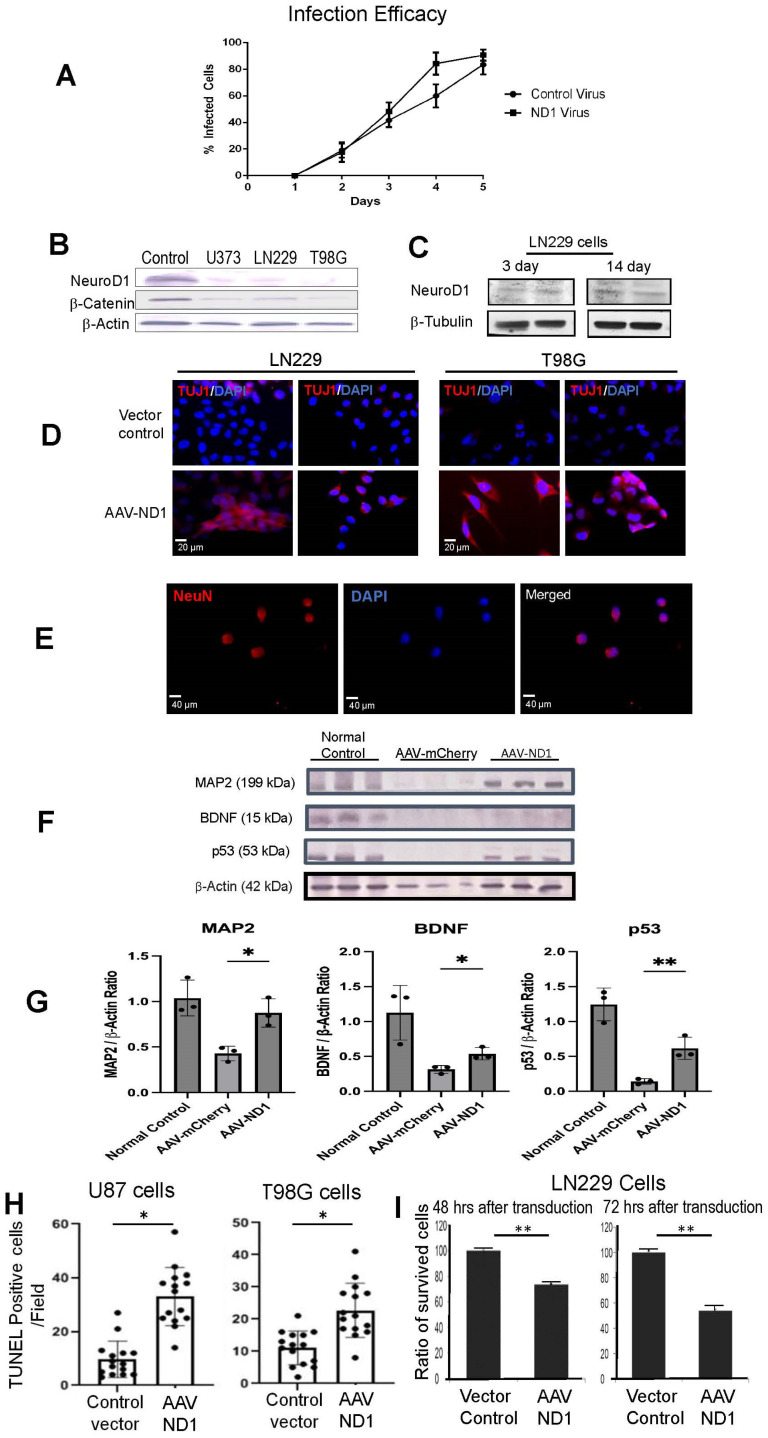
Neuronal transdifferentiation of GBM cells induced by ND1 expression. GBM cells were cultured in a neuron stem cell medium for 10 days before AAV-ND1 transduction. (**A**) Comparisons of GBM cell infection by the control AAV-mCherry vector and AAV-ND1-mCherry vector. Both vectors showed similar infection rates and efficacy in GBM cultures. (**B**) Western blotting revealed a high-level expression of ND1 protein in the normal mouse brain; the ND1 level in GBM cell lines U373, LN229, and T98G was noticeably low or barely detectable. A similar expression pattern was seen with β-catenin. (**C**) AAV-ND1 delivered expression of ND1 in LN229 cells 3 and 14 days after transduction. (**D**) Immunocytochemical staining 7 days after viral delivery detected the neuronal marker TUJ1 (red) in AAV-ND1 transfected cells, no TUJ1 was seen in vector control cells. (**E**) The mature neuronal marker NeuN (red) was observed 21 days after ND1 expression. Blue color was the nucleus marker DAPI. (**F**) Western blot (WB) assays revealed the absence of neuronal marker MAP2, the trophic factor BDNF, and the normal cell marker p53. All these markers significantly increased in ND1-reprogrammed GBM cells. (**G**) Quantification of the WB test in (**F**). N = 3, * *p* < 0.05 vs. control vector’ ** *p* < 0.01 vs. controls; One-way ANOVA. (**H**) The TUNEL assay of U87 and T98G cells confirmed the increased DNA damage and apoptotic cell death of ND1-expressing cells. N = 3, *. *p* < 0.05 vs. control vector, Student unpaired *t* test. (**I**) In the MTT assays, AAV-ND1 converted cultures exhibited poorer survival compared to control vector-treated GBM cultures. N = 3, ** *p* < 0.01 vs. control vector, unpaired *t* test.

**Figure 2 cells-13-00897-f002:**
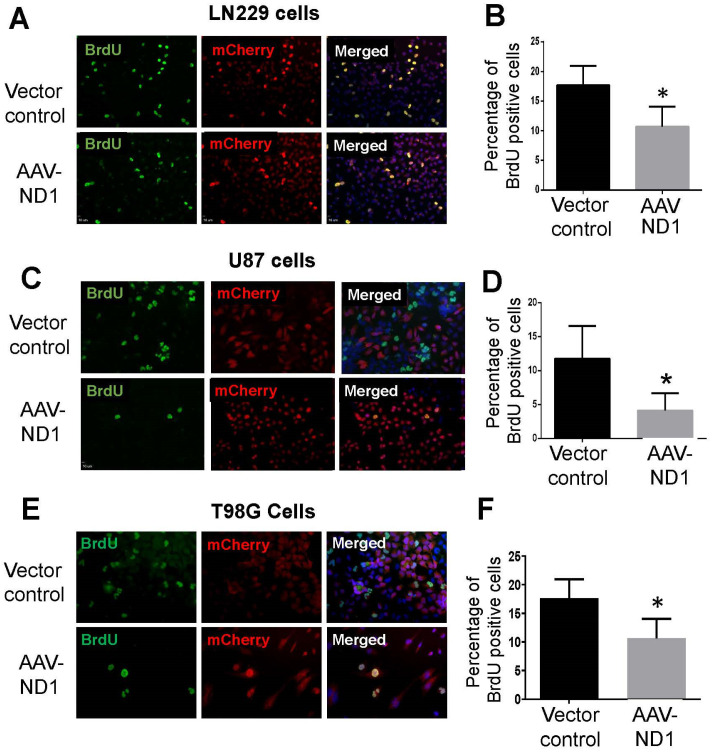
Attenuated proliferation of ND1-reprogrammed GBM cells. Immunocytochemical assays measured the proliferation of GBM cells in different cell lines 3 days after viral transduction. (**A**,**C**,**E**) Representative fluorescent images of three GBM cell lines. Green: BrdU-positive cells; Red: mCherry-labeled transfected cells. (**B**,**D**,**F**) Quantified measurements in the experiments of (**A**,**C**,**E**), respectively. ND1 expression significantly decreased BrdU expression in AAV-ND1 transduced multiple GBM cells. N = 3 cultures, * *p* < 0.05 vs. control.

**Figure 3 cells-13-00897-f003:**
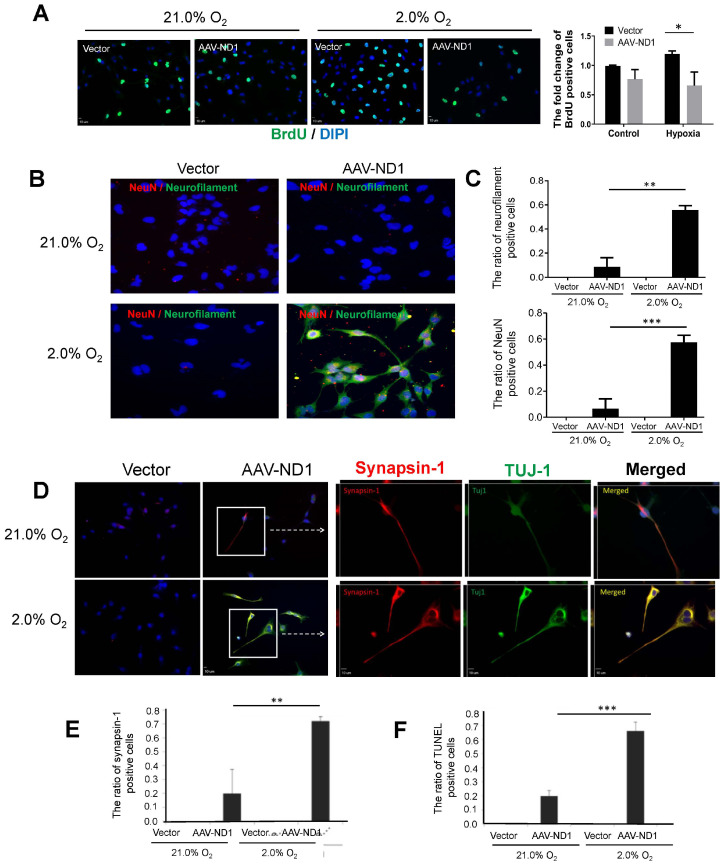
Promoting effects of hypoxia on ND1-induced neuronal conversion of GBM cells. Immunocytochemical assessments 7 days after ND1-transduction of GBM cells performed under 2% and 21% O_2_, respectively. (**A**) Fluorescent imaging of BrdU staining and the qualified analysis showed significant reduction of proliferating GBM cells (U373 cell line) reprogrammed after AAV-ND1. * *p* < 0.05 vs. vector control. (**B**) Imaging analysis demonstrated the promoting effect of hypoxia on the expression of mature neuronal marker NeuN of LN229 cells. Typical neuronal morphology such as prolonged axonal tracks as seen with these cells. (**C**) Quantified analysis of the results in (**B**). ** *p* < 0.01 vs. vector controls; *** *p* < 0.005 vs. vector controls; N = 3. (**D**) The staining of synapsin-1 protein (red) and TUJ-1 (green) in vector control and AAV-ND1 treated U373 cells shown at low and high magnifications. The frame indicates the area of the image with enlarged magnification. (**E**) Quantified analysis of synapsin-1 expression at low and high O_2_ levels. ** *p* < 0.01 vs. vector controls. N = 3. (**F**) TUNEL staining examined cell death 7 days after virus transduction under 2% O_2_ and 21% O_2_ conditions, respectively. The hypoxic condition markedly augmented cell death of ND1-reprogrammed cells. N = 3 assays. *** *p* < 0.005 vs. controls and reprogrammed cells under 21% O_2_.

**Figure 4 cells-13-00897-f004:**
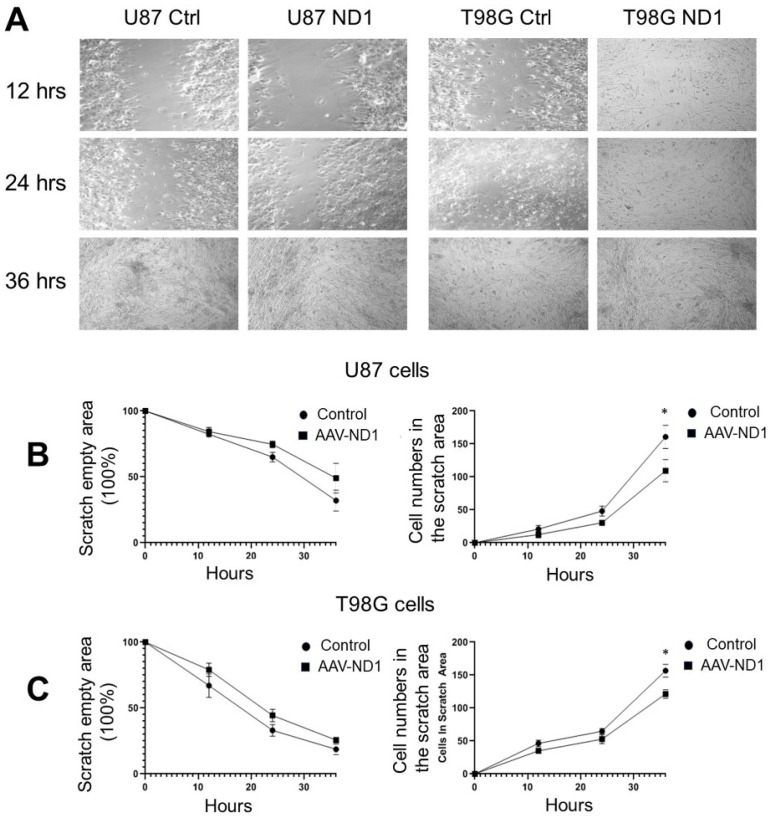
Reduced migrative activity of ND1-converted GBM cells. Wound-healing assay was performed in U87 and T98G cell cultures 3–5 days after ND1-reprogramming. (**A**) Phase contrast images show wounded areas and different times after the formation of the wounded (empty) area in different groups. (**B**,**C**) Time courses of cell migration to cover the empty area and the cell count inside the area at 12, 24, 36, and 48 h after scratch, in U87 cultures (**B**) and G98G cultures (**C**). ND1-expressing cells were significantly fewer inside the empty area compared to the number of migrated vector-controlled cells. * *p* < 0.05 vs. vector controls, N = 3 cultures/group. ANOVA test.

**Figure 5 cells-13-00897-f005:**
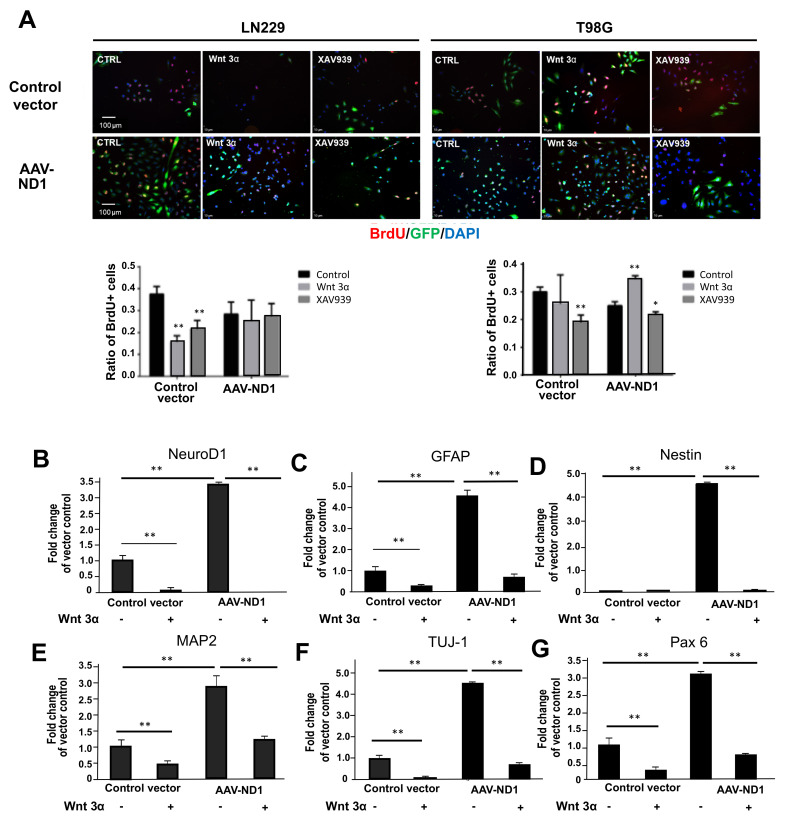
Regulatory roles of the Wnt pathway in ND1-induced GBM reprogramming and cell proliferation. Immunocytochemical staining of ND1 and neuronal markers with the manipulations of the Wnt/β-catenin signaling. (**A**) Cell proliferation of BrdU staining was assessed 48 h after transduction; GFAP and DAPI were used for cell counting with and without Wnt 3α (100 nM) or XAV939 (20 ng/mL) in vector control and AAV-ND1 cultures. Wnt 3α and XAV939 were co-applied with the virus. Bar graphs underneath are the quantified cell counts of BrdU-positive cells, showing significant inhibitory effects of Wnt 3α and XAV939 in LN229 naïve cultures but not in ND1-expressing cultures. In T98G cells, XAV939 showed inhibitory effects in both vector control and ND1 converted cells, while Wnt 3α increased BrdU-positive cells in ND1 cells. * *p* < 0.05 and ** *p* < 0.01 vs. the control vector. (**B**–**G**) Quantified expressions of reprogramming and neuronal markers 7 days after transduction of LN229 cells, including ND1 (**B**), GFAP (**C**), Nestin (**D**), MAP2 (**E**), TUJ-1 (**F**), and Pax 6 (**G**). Among these markers, Wnt 3α displayed strong inhibitory actions in control GBM cultures and ND1 reprogrammed cells. These data suggest that although the Wnt/β-catenin pathway may have variable regulations on proliferation/cell cycle of GBM cells, the expression/activation of Wnt signaling is required for neuronal transdifferentiation of GBM cells. ** *p* < 0.01 vs. the control vector.

**Figure 6 cells-13-00897-f006:**
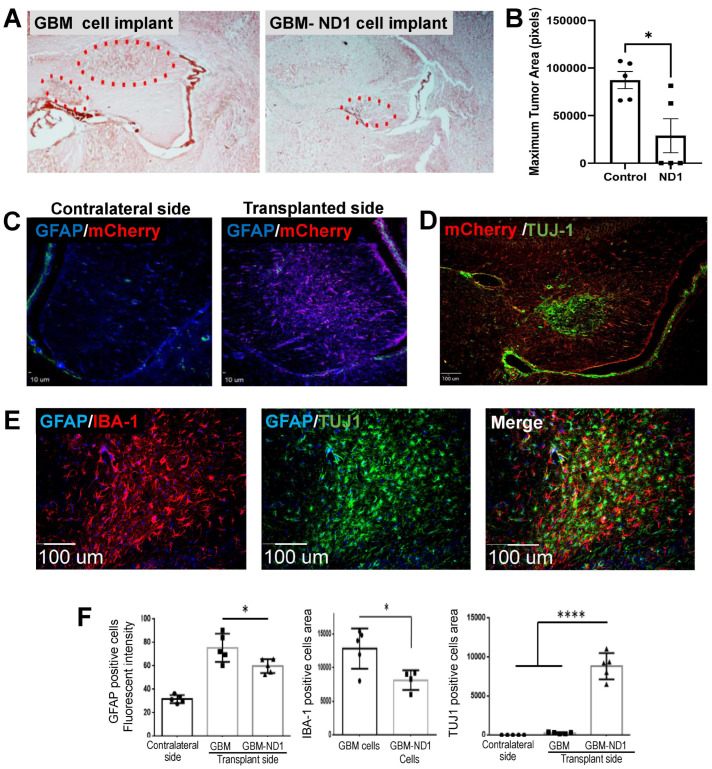
Inhibition of tumor growth by ND1-reprogramming in a GBM mouse model. An orthotopic GBM model was established by implanting U373 cells transduced by control AAV-mCherry or AAV-ND1-mCherry vector into the white matter region of the right hemispheres of adult mice. Tumors were detected as soon as 7–14 days after implantation. (**A**) In the H&E assay, one or more tumors were identified (dotted line) in and around the injection regions that received control GBM cells, while significantly smaller single tumors or no tumor tissue was visible in mice that received ND1-expressing cells. (**B**) Significantly decreased tumor size formed by ND1 cells compared to control cells. N = 5/group, * *p* < 0.05 vs. controls. (**C**) Immunohistochemical staining of GFAP (blue) and mCherry (red) in the contralateral and transplantation sides of the brain, confirming the seeding of viral transduced cells in the targeted brain region. (**D**) TUJ-1 staining (green) and mCherry (red) revealed converted neuronal cells in the white matter area. (**E**) Accumulations of IBA-1positive microglia/microphage (red) in the injection site. In the same region, there were abundant TUJ-1-positive cells (green) indicative of ND1-converted neurons that did not overlay with GFAP (blue) or IBA-1 (red) fluorescence. (**F**) Quantified analyses of GFAP, IBA-1, and TUJ-1 positive cells. Please note there was basically no TUJ-1 fluorescence in the contralateral side and the control GBM tumor. A drastic increase of TUJ-1 expression was only seen with ND-1 cell transplantation. N = 5–6 mice/group, * *p* < 0.05 vs. controls, **** *p* < 0.001 vs. controls.

## Data Availability

A poster was presented in the Annual Meeting of Society of Neuroscience 2023: Estaba, T., Jiang, M.Q., Dharanendra, S., Gu, X., Wu, A., Berglund, K., Yu, S.P. Wei, L.; Reprogramming of Astroglioma Cells for Redirecting the Tumor Cell Fate as a Potential Anti-Cancer Gene Therapy. Program No. PSTR392.02. 2023 Neuroscience Meeting Planner. Washington, D.C.: Society for Neuroscience, 2023. Online. Presentation Number: PSTR392.02.

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
