# Peer review of "Reprogramming Glioblastoma Cells into Non-Cancerous Neuronal Cells as a Novel Anti-Cancer Strategy"

_cells, 2024, doi:10.3390/cells13110897_

Round 1

Reviewer 1 Report

Comments and Suggestions for Authors

The article titled "Direct Reprogramming and Conversion of Glioblastoma Cells into Non-Cancerous Induced Neurons as a Potential Anti-Cancer Therapy" makes a substantial contribution to the field of cancer therapy and genetic reprogramming by introducing a new method for treating a complex disease. Nevertheless, there are aspects where the presentation and organization of the content could be improved:

1. The title, while descriptive, could be simplified for better clarity and impact. For example, "Reprogramming Glioblastoma Cells into Neurons: A Novel Anti-Cancer Strategy"

2. The abstract is thorough but somewhat lengthy and complex. Improving conciseness would enhance the readability and impact of the text. The key findings and their implications could be emphasized more clearly in order to provide immediate attention.

3.  The abstract mentions the use of ND1 for direct reprogramming but does not sufficiently highlight the novelty or the potential impact of this research compared to existing therapies. Emphasizing how this approach could offer advantages over current treatments would strengthen the abstract.

4. The text switches between "ND1" and "NeuroD1" without consistency. Choose one term and stick with it throughout the document to avoid confusion.

5. Certain claims, such as specific gene expressions and their impacts, are made without specified citations. 

6. The methods section could be improved by adding more subheadings.

7. The transformation of GBM cells into cells with neuronal markers and reduced tumorigenic potential holds significant promise. The authors should elaborate on how these transformed cells might interact with the surrounding brain environment. Additionally, discussing potential strategies for integrating these cells into existing treatment protocols could provide critical context and insights into practical applications.

8. More elaboration on the potential challenges of translating these findings into clinical applications would benefit the discussion. Specifically, it would be helpful if the authors could address concerns regarding the delivery mechanisms for ND1, the possibility of off-target effects, and the long-term stability and safety of the reprogrammed cells.  

Comments on the Quality of English Language

The overall English and grammatical structure of the text are commendable; however, there are several areas where enhancements could be made to improve clarity and readability.

Author Response

Reviewer 1

Comments and Suggestions for Authors

The article titled "Direct Reprogramming and Conversion of Glioblastoma Cells into Non-Cancerous Induced Neurons as a Potential Anti-Cancer Therapy" makes a substantial contribution to the field of cancer therapy and genetic reprogramming by introducing a new method for treating a complex disease. Nevertheless, there are aspects where the presentation and organization of the content could be improved:

  1. The title, while descriptive, could be simplified for better clarity and impact. For example, "Reprogramming Glioblastoma Cells into Neurons: A Novel Anti-Cancer Strategy"

Thanks to the constructive suggestion, the title of the paper has been modified accordingly.

  1. The abstract is thorough but somewhat lengthy and complex. Improving conciseness would enhance the readability and impact of the text. The key findings and their implications could be emphasized more clearly in order to provide immediate attention.

The Abstract has been revised with more focus on the results and impacts.

  1. The abstract mentions the use of ND1 for direct reprogramming but does not sufficiently highlight the novelty or the potential impact of this research compared to existing therapies. Emphasizing how this approach could offer advantages over current treatments would strengthen the abstract.

The novelty of this investigation has been highlighted in the title, abstract and text of the paper. Advantages and future directions of the reprogramming strategy have been discussed.

  1. The text switches between "ND1" and "NeuroD1" without consistency. Choose one term and stick with it throughout the document to avoid confusion.

This has been corrected.

  1. Certain claims, such as specific gene expressions and their impacts, are made without specified citations. 

Additional papers have been cited in several locations.

  1. The methods section could be improved by adding more subheadings.

Additional subheadings were added into the Methods section.

  1. The transformation of GBM cells into cells with neuronal markers and reduced tumorigenic potential holds significant promise. The authors should elaborate on how these transformed cells might interact with the surrounding brain environment. Additionally, discussing potential strategies for integrating these cells into existing treatment protocols could provide critical context and insights into practical applications.

We have discussed the possible use of converted cells as an additional benefit of the reprogramming strategy. Since current data lack the functional verification of neuronal activities such as the firing of action potentials, it appears too early to discuss how they can integrate into neuronal networks. In the discussion, we pointed out the advantages of converting GBM cells into non-functional neurons as an anti-cancer treatment and fully functional neurons for additional purpose of brain repair.

  1. More elaboration on the potential challenges of translating these findings into clinical applications would benefit the discussion. Specifically, it would be helpful if the authors could address concerns regarding the delivery mechanisms for ND1, the possibility of off-target effects, and the long-term stability and safety of the reprogrammed cells.  

 Potential challenges and the delivery methods for translational applications are discussed. We also recognize that future investigations will be needed for evaluate possible reprogramming of GSCs and any side-effects. 

 Comments on the Quality of English Language

The overall English and grammatical structure of the text are commendable; however, there are several areas where enhancements could be made to improve clarity and readability.

English writing was checked, and improvements were made in the revised paper.

Reviewer 2 Report

Comments and Suggestions for Authors

 The Wnt experiments of Fig.5 are not convincing:

 Fig5A. If  Wnt activates neurogenesis and ND1 acts via this pathway, then in their context Wnt inhibition should abrogate  the  effect of ND1. However, their results are conflicting:

1. Proliferation:  a. In LN229 either Wnt3 or the inhibitor lowered the proliferation  of control and  had no effect in ND1-expressing cells. b. In T98G  results are more clear: the inhibitor only reduced proliferation and ND1 expressing cells  seems to be benefited by Wnt and inhibited by XAV.

2.  Fig5B-G.  a. Which GB is the one shown. LN or TG98  ?

b. If ND1 mimics Wnt  why  Wnt3 reduces all the genes studied ? I would expect that this is the effect of the XAV and not the Wnt3.

c. Why the endogenous ND1 expression and even more the AAV –mediated ND1 expression is reduced by Wnt3?

 Other comments 1.  They must use and independent assay to measure Wnt signalling activity in their experiments, i.e. catenin location/abundance or target gene activation.

2. What is the significance of the genes they showed to be down regulated?  It seems that some are tumor-supressors while others are pro-oncogenic and highly expressed in   stem or aggressive cellsbut they are all down regulated by Wnt3.

2. The role of Wnt in gliomas must be criticall  discussed  and in more detail. Many sources support a GB-promoting role of an hyperactive Wnt  pathway (doi.org/10.1038/labinvest.2015.140).

Conclusion: They must clarify the above role of Wnt  or remove the data relevant to  Fig5.

Comments on the Quality of English Language

minor editing

Author Response

Reviewer 2 (please note a figure in the response is not shown in this note, you can see the figure in the attached Word file)

Comments and Suggestions for Authors

 The Wnt experiments of Fig.5 are not convincing:

 Fig5A. If  Wnt activates neurogenesis and ND1 acts via this pathway, then in their context Wnt inhibition should abrogate the effect of ND1. However, their results are conflicting:

We appreciate the reviewer brought out this important issue. We have discussed that because of the complicated, sometimes opposing regulatory roles of the Wnt pathways, Wnt signaling not only promotes neurogenesis, activities of Wnt pathways also play important roles in keeping cell stemness status and promote GBM proliferation, both are opposite to neuronal differentiation. We have discussed that it is likely the balance of different regulations of Wnt signaling would determine the ultimate outcome of increasing and decreasing Wnt signaling. This can especially be true with different GBM cell lines that have distinct gene expression profiles and possible mutated genes. We agree that the exact explanation for the contradictory results is unknown. However, we feel it is important to report “negative” or “inconsistent” data in research papers as long as they are from reliable experiments.   

  1. Proliferation:  a. In LN229 either Wnt3 or the inhibitor lowered the proliferation  of control and  had no effect in ND1-expressing cells. b. In T98G  results are more clear: the inhibitor only reduced proliferation and ND1 expressing cells  seems to be benefited by Wnt and inhibited by XAV.

See discussion above and in the revised test.

  1. Fig5B-G.  a. Which GB is the one shown. LN or TG98?

LN229 cells were shown here. The cell line name has been specified.

  1. If ND1 mimics Wnt  why  Wnt3 reduces all the genes studied ? I would expect that this is the effect of the XAV and not the Wnt3.

We are sorry for the confusion. This investigation does not suggest ND1 mimics Wnt signalling. As discussed and shown in many studies, Wnt represents multifaceted, complex regulations with broader cellular and molecular consequences. We checked original record of these experiments and confirmed the effects were from Wnt3α but not XAV939. Currently, Wnt’s regulation on transdifferentiation from GBM to neuron is a new area and has not been investigated before. It may involve even more coordination of the Wnt pathways.  

  1. Why the endogenous ND1 expression and even more the AAV –mediated ND1 expression is reduced by Wnt3?

We agree that there is no readily available explanation for this observation, except that Wnt appears to have an overall inhibitory effect on ND1-induced neuronal transdifferentiation.   

Other comments 1.  They must use and independent assay to measure Wnt signalling activity in their experiments, i.e. catenin location/abundance or target gene activation.

We appreciate this constructive suggestion. The submitted paper is an initial effort to report on the efficacy of innovative reprogramming of GBM cells. We have continued the investigation by more closely examining the underlying mechanism by which the ND1-induced transdifferentiation was uniquely regulated. Below is a figure showing three GBM cell lines and distinctive results from increased Wnt signaling. We hope more information will be available in a following report to reveal a better understanding of the observation.

  1. What is the significance of the genes they showed to be down regulated?  It seems that some are tumor-supressors while others are pro-oncogenic and highly expressed in stem or aggressive cells but they are all down regulated by Wnt3.

The reduced expression of MAP2, TUJ1 and PAX6 is highly consistent with the low expression of ND1 in these cells, supporting attenuated neuronal differentiation. PAX6 has been used as an astrocyte marker; it also acts as an oncogene responsible for induction of cancer stemness properties (Ooki et al., Epigenetically regulated PAX6 drives cancer cells toward a stem-like state via GLI-SOX2 signaling axis in lung adenocarcinoma. Oncogene. 37:5967-5981, 2018). The reduction of PAX6 may suggest a reduced GSC phenotype; again, the gene expression profile may be affected by a comprehensive interactions of multiple signalling pathways during the course of transdifferentiation.

  1. The role of Wnt in gliomas must be criticall  discussed  and in more detail. Many sources support a GB-promoting role of an hyperactive Wnt  pathway (doi.org/10.1038/labinvest.2015.140).

We have reviewed that hyperactivity of Wnt pathway is a characteristic feature of GBM tumors. On the other hand, activation of Wnt signalling is a strong promoting factor of neuronal differentiation. Thus, Wnt signalling plays distinctive roles in these cell phenotype regulations; which theoretically are contradictory to each other in the ND1-induced transdifferentiation. As a novel strategy and manipulation of cellular fate, little is known about the molecular regulation of this type of cell lineage change.  

Conclusion: They must clarify the above role of Wnt  or remove the data relevant to  Fig5.

As discussed above, we believe that all scientific research and data reports should be based on experimental evidence, even if the observation deviates from conventional opinions. Since we trust the result in Figure 5 is a true presentation of experimental results, it is important to communicate these unexpected data to others. Because of the novelty of the strategy and the complicated nature of Wnt regulations, it is clear that much more is needed to clarify the role of Wnt in the transdifferentiation of GBM cells, which likely depends on multiple conditions such as cell type, gene profiles, involvement of different Wnt pathways and interaction with other signaling.

Comments on the Quality of English Language

minor editing

The manuscript has been extensively revised and English has been improved.

Round 2

Reviewer 2 Report

Comments and Suggestions for Authors

I have no commends.